# Role of Quality of Life as Endpoint for Inflammatory Bowel Disease Treatment

**DOI:** 10.3390/ijerph18137159

**Published:** 2021-07-04

**Authors:** Cristina Calviño-Suárez, Rocío Ferreiro-Iglesias, Iria Bastón-Rey, Manuel Barreiro-de Acosta

**Affiliations:** IBD Unit, Gastroenterology Department, University Hospital of Santiago de Compostela, 15706 Santiago de Compostela, Spain; criscalvino.su@hotmail.com (C.C.-S.); rocioferstg@hotmail.com (R.F.-I.); iria.baston@gmail.com (I.B.-R.)

**Keywords:** quality of life, Crohn’s disease, ulcerative colitis

## Abstract

Inflammatory bowel diseases (IBDs) are chronic disabling conditions, characterized by an unpredictable course with flare-ups and periods of remission, that frequently affect young people and require lifelong medical follow-up and treatment. For years, the main endpoints of IBD treatment had been clinical remission and response, followed by biomarker normalization and mucosal healing. In the last decades, different therapies have been proved to be effective to treat IBD and the use of patient reported outcome (PRO) have become more relevant. Therefore, health-related quality of life (HRQoL) that has been defined as the value assigned to the duration of life influenced by physical and mental health, has been suggested as an important endpoint for IBD management since multiple studies have shown that IBD impairs it, both physically and psychologically. Thus, HRQoL has been included as an outcome in numerous studies evaluating different IBD therapies, both clinical trials and real-life studies. It has been assessed by using both generic and specific disease tools, and most treatments used in clinical practice have been demonstrated to improve HRQoL. The relevance of HRQoL as an endpoint for new drugs is going to increase and its management and improvement will also improve the prognosis of IBD patients.

## 1. Introduction

Traditional medicine was focused on the physical side of the illness and death rates and life expectancy were the main measures used to evaluate people’s health. This excludes the fact that, in most diseases, the state of health is deeply influenced by mood, coping mechanisms to different situations and social support. The higher prevalence of chronic conditions, as a consequence of the decline of infectious diseases, as well as the development of new technologies that reduced pain, have made necessary newer and more sensitive outcomes beyond morbidity and biological functioning [1]. Quality of life (QoL) has been considered as a component of health since 1947 when the World Health Organization (WHO) began to define health not only as the absence of disease, but also as a state of physical, mental and social well-being [2]. Health-related quality of life (HRQoL) has been defined as the value assigned to the duration of life influenced by health, which is modified by impairments, functional state, perceptions and opportunities that are in turn influenced by diseases, injury and treatments [3]. HRQoL only includes components that are part of an individual’s health and, therefore, excludes other aspects of QoL, as political or economic factors [4].

Inflammatory bowel diseases (IBD) are chronic, progressive and disabling conditions affecting young people that have a negative impact on their HRQoL [5]. For years, the main endpoints of IBD treatment had been clinical remission and response. Afterwards, new targets like biomarkers and mucosal healing have been introduced in new drug evaluations and, in the last decades, the use of patient-reported outcome (PRO) has also become especially important. In 2015 the Selecting Therapeutic Targets in Inflammatory Bowel Disease (STRIDE) program was initiated by the International Organization for the Study of Inflammatory Bowel Diseases (IOIBD). It examined potential treatment targets for IBD to be used for a “treat-to-target” clinical management strategy using an evidence-based expert consensus process. In these first recommendations, improvement of HRQoL was only suggested as part of PRO [6]. In the recently published STRIDE II consensus, HRQoL has more weight and it is recommended as an important endpoint for IBD management [7].

## 2. How Can We Measure HRQoL in IBD?

There are two main types of HRQoL tools to evaluate patients with IBD: disease-specific and generic. Disease-specific tools evaluate symptoms and compare the effect of different treatments, while generic tools allow for comparisons between different population and illnesses.

It is important to take into account two psychometric considerations for choosing which instrument to measure HRQoL:− Reliability is the probability that a questionnaire will perform its intended function adequately. A reliable measure is one that provides consistent and accurate information.− Validity is how accurately a method measures what it is intended to measure. A tool is valid when it measures the characteristic that it claims to measure.

Plenty of IBD-specific HRQoL tools have been developed and validated for IBD patients [8]. Nevertheless, the majority of these instruments have had no patient involvement in their development [9]. In Table 1, we summarize the main characteristics of the most widely used tools and some other options designed for specific IBD cohorts.

The Inflammatory Bowel Disease Questionnaire 32 (IBDQ-32) and Inflammatory Bowel Disease Questionnaire 36 (IBDQ-36) are the most commonly used [10]. IBDQ-32 is a 32-item questionnaire that has been demonstrated to be reliable and valid. It includes four aspects of the patients’ life and the main domains are intestinal symptoms (10 items), systemic symptoms (five items), social (12 items) and emotional domains (five items). Häuser et al. conducted a validation study of the German version of the IBDQ (IBDQ-D) for patients with ileal pouch anal anastomosis (IPAA) for UC, and they observed that it was a reliable tool in this setting although it had some limitations in terms of validity [11]. The short version of IBDQ-32 is the Short Inflammatory Bowel Disease Questionnaire (SIBDQ). SIBDQ also contains symptom, social and emotional sections. IBDQ-36 is a 36-item questionnaire that has also been proven to be valid and reliable. It comprises the following points: intestinal symptoms (eight items), systemic symptoms (seven items), social (6 items) and emotional domains (eight items), and functional impairment (seven items). The short version of IBDQ-36 is IBDQ9. IBDQ9 only contains one domain (total score) and the comprehensiveness is lower than IBDQ-36 [9].

Another tool, Crohn’s Life Impact Questionnaire (CLIQ), composed of 27 dichotomous items, is focused on how the impairments affect need fulfilment. It has demonstrated good validity and reproducibility, and it is easy to complete in a few minutes [12]. Recently, the Crohn’s Anal Fistula Quality of Life (CAF-QoL) has been developed to evaluate the impact of anal fistula. It is a new PRO measure for Crohn’s perianal fistula that has been validated. CAF-QoL is a 28-item questionnaire that has demonstrated to be internally consistent, reliable, stable and valid [13]. Among them, the best questionnaires related to relevance, comprehensiveness and comprehensibility are IBDQ-32 and CLIQ. In Table 1, we summarize the main characteristics of the most widely questionnaires used.

Other examples of disease-specific instruments are the Crohn’s and Ulcerative Colitis Questionnaire (CUCQ), Inflammatory Bowel Disease Questionnaire 30 (IBDQ-30), Norwegian Inflammatory Bowel Disease Questionnaire (IBDQ-N), Cleveland Global Quality of Life (CGQL), Short Health Scale (SHS), Edinburgh Inflammatory Bowel Disease Questionnaire (EIBDQ), short Inflammatory Bowel Disease Questionnaire 10 (sIBDQ-10) and Inflammatory Bowel Disease Disability Index (IBD-DI). In paediatric IBD patients, the IMPACT series tools (IMPACT, IMPCT-II and IMPACT III) were used to evaluate the HRQoL. IMPACT was proven to be valid and contains 4 domains: symptoms, physical, emotional and social domains [8,14].

The generic questionnaires most commonly used are the Generic 36-item Short Form Survey (SF-36) and The EuroQoL–dimension (EQ-5D). SF-36 was developed in the USA for use in the Medical Outcomes Study (MOS). It is a generic scale that provides quantitative information related to HRQoL and has good validity and reliability. It is frequently reported as two separate figures, a physical component score (PCS) and a mental component score (MCS), which included a total of 36 items allocated in eight domains: physical functioning (10 items), role physical (four items), social functioning (two items), bodily pain (two items), mental health (five items), role emotional (three items), general health perceptions (five items) and one item about general health [15]. EQ-5D is a generic, reliable and valid instrument developed by the EuroQoL group. It can be used to assess HRQoL but also the cost-utility analysis of health care interventions [16]. Other similar instruments can be World Health Organization Quality of Life (WHOQOL)-BREF, Short Form SF-12, Satisfaction with Life Scale (SWLS), EORTC Quality of Life Questionnaire C-30, Quality of Well Being Scale or Health Utilities Index [17,18]. In paediatrics, the generic tools more widely used are PedsQ1, Child Health Questionnaire (CHQ), KINDL, KINSCREEN 27, DISABKIDS HRQOL [19].

## 3. Quality of Life Studies in UC

To date, multiple studies have reported that UC impairs QoL, which can also be affected by demographic, psychological and socioeconomic factors [20,21]. Clinical activity was pointed out as the factor with the most negative impact in HRQoL [22], although it has been shown that it’s still compromised during quiescent disease as compared to the general population [23]. Rasmussen et al. observed that bowel frequency, urgency and rectal bleeding are the symptoms that most significantly affect these patients’ HRQoL [24]. Apart from the physical symptoms, IBD patients complain about an important emotional burden which is barely addressed during follow-up appointments [25,26]. Different therapies have been proved to be effective to treat UC and improve the HRQoL of those who suffer from it (Table 2) although its administration schedules and side effects can also negatively affect HRQoL.

Since the 1990s, several studies were conducted to assess the effect of 5-ASAs in UC patients HRQoL. Back then, Robinson et al. published a randomized, double-blind, multicentre trial which included 374 UC patients under oral mesalamine 1 g, 2 g or 4 g daily versus placebo. They evaluated twelve HRQoL parameters (five symptoms and seven aspects of general life) and observed a significant improvement of all of them in 2 g and 4 g daily mesalamine groups versus placebo [27]. Probert et al. undertook a clinical trial where 127 patients with extensive mild-to-moderately active UC were randomized to a combined oral and rectal mesalazine or oral mesalazine with rectal placebo. The EQ-5 D questionnaire was used to evaluate HRQoL at 2, 4 and 8 weeks. Although there were no differences between both groups at baseline and 8 weeks, they observed significant improvements at week 4 in the combination therapy group in ‘morbidity´, ‘usual activity’ and ‘anxiety/depression’ domains, which reflects a quicker efficacy of combined oral and rectal treatment [28].

Only a few studies have been undertaken to evaluate the impact of corticosteroids in UC patients’ HRQoL. In an IBD cohort study, the IBSEN study group didn’t encounter any significant difference between UC patients treated with corticosteroids compared to non-users after a five-year disease course [29]. However, seven years later, the same group observed that the use of corticosteroids leads to a worsening HRQoL in UC patients [20]. On the same line, corticosteroids were pointed out as the only treatment between the top ten factors with the strongest impact on HRQoL [30].

The evidence on the effect of immunomodulators in UC patients’ HRQoL is scarce. Neither has any difference been found between azathioprine users and non-users in terms of HRQoL in previously mentioned UC group of the Norwegian cohort [29]. Nonetheless, this drug has been associated with a HRQoL improvement in an English IBD patients survey from 2007 [31] and a more recent Saudi IBD cohort study, whereby those patients treated with azathioprine registered higher EQ-5D scores (β = 9.35; 95% CI: 0.486–18.22; *p* = 0.003) [32].

Several studies have reported the biological treatments’ influence on UC patients’ HRQoL. As infliximab was the first biological therapy approved for UC in 2005, the ACT 1 and ACT 2 were the first trials demonstrating higher IBDQ scores from those patients treated with 5 or 10 mg/kg of infliximab when compared to placebo at week 8 (mean score of 40, 36 and 28, respectively; *p* < 0.001). Patients on infliximab maintenance therapy kept this HRQoL improvement at weeks 30 and 54, also observed on the SF-36 score [33]. Recently, it has been shown that infliximab still improves UC patients’ HRQoL in a prospective study in which a significant change of IBDQ score was seen from 116.20 at baseline to 176.62 at week 54 (*p* = 0.02) [34].

In 2012, adalimumab was proved to be effective as an induction and maintenance treatment for UC in the ULTRA 1 and ULTRA 2 trials, and it has also been associated with a better HRQoL among anti-TNFα naïve patients in comparison to placebo at weeks 8 (mean IBDQ score of 48 vs. 31, *p* = 0.039) and 52 (mean IBDQ score 102 vs. 75, *p* = 0.004) [35]. Afterwards, these results would be confirmed in real-life studies as in the InspirADA study whereby patients with moderate to severe UC on adalimumab reported a significant improvement in their HRQoL, which was assessed by the SIBDQ (mean change ± SD: 17.4 ± 14.5) and the EQ-5D (index: 0.1 ± 0.2; VAS 19.5 ± 25.8) [36].

In a post-hoc analysis from PURSUIT-SC induction trial, the highest HRQoL scores at week 6 were communicated by golimumab treated patients when compared to placebo (IBDQ 27.2 vs. 14.6, *p* < 0.001; SF-36 PCS 4.14 vs. 2.46 and MCS 4.89 vs. 1.60, *p* < 0.01 for both comparisons). There haven’t been any significant differences found in the mean mentioned HRQoL scores between the two doses of golimumab (400/200 mg vs. 200/100 mg) [37].

Apart from anti-TNFα, vedolizumab and ustekinumab have also proved to be effective for improving UC patients’ HRQoL. Results from the GEMINI 1 trial revealed a significant improvement on IBDQ and EQ-5D scores on vedolizumab every 8 or 4 weeks as compared to placebo at week 52 (IBDQ mean difference ≥ 21.1 and EQ-5D mean difference ≥ 9.3 for both vedolizumab groups). However, this significant difference on EQ-5D was just detected on the vedolizumab every 4 weeks group [38]. Recently, the VARSITY trial, the first head-to-head trial comparing biological treatments for UC, reported results (mean IBDQ score ± SD) in favor to vedolizumab in terms of HRQoL improvement at week 30 (61.3 ± 39.8 vs. 52.6 ± 42.8) and 52 (66.1 ± 41.8 vs. 60.4 ± 42.2) [39]. On the other hand, UNIFI trial showed that ustekinumab raises the HRQoL after induction in comparison with placebo, and this benefit was sustained at week 44 [40] and 92 on the long-term extension study on which 55.6% of patients who had been treated with ustekinumab were in IBDQ remission [41].

Tofacitinib is the first small molecule approved as a treatment for UC and it also has been shown to achieve greater changes in IBDQ and SF-36 scores versus placebo. In OCTAVE induction 1 and 2, mean IBDQ changes from baseline to week 8 was 40.7 and 44.6 with tofacitinib 10 mg twice daily versus 21 and 25 with placebo, respectively (*p* < 0.001). Mean SF-36 changes were comparable with the IBDQ changes and both were sustained at week 52 [42].

Regarding surgery related to UC, Pica et al. undertook a cohort study to compare HRQoL between UC patients under medical treatment with some others who underwent total colectomy with ileorectal anastomosis (IRA) or IPAA [43]. About 63% patients in the medical treatment and IRA groups were in remission, whilst 47.9% patients in the IPAA group reported complications. Their HRQoL survey was composed of four domains (intestinal symptoms, systemic symptoms, emotional and social function), and the only significant difference they found was a worse intestinal-symptoms score in the IRA group. Likewise, no differences were found in the median IBDQ score between IPAA, ileostomy and anti-TNFα treated patients in a later Dutch study (183, 181 and 181, respectively, *p* = 0.27). Nevertheless, it was noted that IPAA patients referred further HRQoL deterioration due to bowel symptoms in comparison with the another two groups (*p* ≤ 0.01) [44]. Similarly, a Belgium cohort study showed worse scores reported by IPAA patients compared with anti-TNFα users for ´antidiarrheal medication use´, ´stool frequency´ and ´perianal skin irritation frequency´ domains (*p* < 0.001 for all comparisons). However, it was observed significantly higher general health scores (SF-36) from IPAA patients (*p* = 0.042) and no differences were detected on EQ-5D nor IBD-DI between both groups [45].

## 4. Quality of Life Studies in CD

There are several studies that have shown an association between CD and significant disability and impaired HRQoL. There is a systematic review and meta-analysis that compared HRQoL between CD and UC including physical (2375 participants) and mental scores (2664 participants). The HRQoL scores were shown to be lower in patients with CD compared with UC, but these differences were borderline significant [46].

In the systematic review of Van der Have et al. including 5735 patients with CD, the HRQoL was consistently impaired by the occupational disability, number of flares, disease activity, need for hospital admission and use of corticosteroids. Furthermore, the biological treatment had a beneficial impact in the HRQoL. The majority of the studies included in this review employed both generic and disease-specific HRQoL measures, the IBDQ and the SF-36 being the most commonly used [47].

Despite the thinking that HRQoL is mainly related with clinical activity in CD, in a study performed in 92 CD patients in remission, scores on the SF-36 were lower than in the general population of similar age and sex. Age, colonic location and previous surgery was related with worse HRQoL [48].

The effectiveness of different treatments improving HRQoL in patients with CD has been evaluated in clinical trials of new drugs, but also real-life studies (Table 2). Only a few studies have been conducted to assess the impact of thiopurines in HRQoL. A prospective study that included 92 IBD patients (68 CD) who started thiopurines showed a significant impairment HRQoL at week 0 with a basal median IBDQ score of 4.99 (range 2.37–6.84) as compared to patients in remission. In the first year after starting treatment, all dimensions of the IBDQ demonstrate a statistically significant improvement that was more pronounced in those patients receiving steroids at the beginning of the study [49]. In a case-control study it was shown a restoration of HRQoL in patients with CD in remission under thiopurines without differences with healthy controls [50].

The impact of anti-TNF in HRQoL in patients with CD is supported by strong evidence based on prospective randomized and real-life studies. In the ACCENT I trial, the authors assessed the effect of infliximab treatment on HRQoL. At the end of the study, at week 54, IBDQ and SF-36 scores in the groups with infliximab demonstrated a substantial improvement. The mean change in the IBDQ at week 54 compared to baseline was 22.1 (*p* = 0.05) in the 5 mg/kg and 30.2 (*p* = 0.001) in 10 mg/kg infliximab maintenance group while it was 8.9 in the placebo group [51].

Other studies confirmed the improvement in HRQoL with infliximab in clinical practice. A retrospective study that included 94 patients with CD who started treatment with infliximab due to moderate-severe active disease, demonstrated an early restoration of HRQoL in 51 patients defined as the overall score of the IBDQ-36 equal or greater than 209 points at week 14 after starting therapy. This early recovery of HRQoL was associated with clinical remission through week 52 [52]. An observational study with 49 patients with CD on infliximab and azathioprine followed for 4 years showed a stable IBDQ-36 in patients on remission with this treatment [53].

In the CHARM study and the open label extension (ADHERE), among 328 patients on adalimumab, more than 50% achieve IBDQ ≥ 170 at 3 years from baseline [54]. In a sub-analysis of this study focused on patients with fistulizing disease (*n* = 48) IBDQ remission was achieved in 60% at 2 years and 52% at 3 years from baseline [55]. In the CARE trial, 945 patients with CD treated with adalimumab induction and maintenance were evaluated in terms of HRQoL and work productivity. The study included patients who were naïve to biologics and patients who failed infliximab. The mean changes in SIBDQ scores from baseline to weeks 4 and 20 were both statistically significant and they were more pronounced in naïve patients. At Week 20, 64% of naïve patients and 55% of infliximab non-responders achieved substantial clinical improvement in total activity impairment [56]. The results of the CHOICE trial in 673 patients with CD who started adalimumab support the findings of the CARE study. Clinically meaningful changes in mean SIBDQ total scores at week 24 were seen in naïve patients as well as in the subgroup of infliximab non-responders [57]. In clinical practice, Saro et al. observed, in a prospective study with 126 CD patients, a significant increase in the IBDQ in the first year after starting adalimumab [56.7 (51.6–61.5) to 67.5 (60.1–73.6)] *p* < 0.05 [58].

The HRQoL in patients with CD under vedolizumab was analyzed in the GEMINI 2 and GEMINI long-term safety (LTS) trial. The mean change from baseline for IBDQ and for EQ-5D were > 51 and > 23 respectively, in the group under vedolizumab representing a clinically meaningful improvement. The improvement in HRQoL was slightly higher in TNF antagonist-naïve patients than for patients with previous failure to anti-TNF, but the differences were not significant [59]. In relation to real-life data, in an observational study that included 21 patients with CD who started vedolizumab an increase of 8.5 points in IBDQ at week 14 from baseline has been shown [60]. At week 52, in a Swedish observational study that included 169 patients with CD on vedolizumab, Eriksson et al. identified a significant reduction in the short health scale [61].

The impact of ustekinumab in HRQoL in patients with CD was evaluated in the UNITI trials. In these studies, patients completed IBDQ at baseline and week 8, 20 and 44. An improvement of ≥16 points in IBDQ score at week 8 was achieved in 68.1% of anti-TNF naïve patients and 54.8% of patients with previous failure to antiTNF. In the maintenance study this improvement was reached in 67.9% of patients under ustekinumab 90 mg q8w at week 44 but only 9.5% achieved HRQoL normalization (IBDQ ≥ 210 points) [62]. A real-life study with ustekinumab in 33 CD patients showed a normalization in the IBDQ at week 52 in 18% of patients. [63].

The effect of intestinal resection in HRQoL was evaluated in many studies. Although surgery remains an important preoperative concern, in the immediately postoperative period, patients generally experience a significant improvement in their HRQoL [64]. In a systematic review including 1108 patients with CD who underwent intestinal resection, HRQoL improved from two weeks after surgery and has remained stable in the long-term [65]. In contrast there are some studies that showed a long-term decrease in the HRQoL of CD patients after surgery [66]; postoperative recurrence, obstructive episodes, the need of new surgery and the number of stools per day were some of factors that contributed to worsening HRQoL [67].

**Table 2 ijerph-18-07159-t002:** IBD treatments that have been shown to improve HRQoL.

Treatment	Study	Measurement Tool(s)	No. Patients	HRQoL: Primary Outcome	Results
5-ASA	Robinson et al. [27]	5 disease-specific and 7 general items	374 UC	Yes	Mesalamine 2 g and 4 g daily was significantly superior to placebo in improving each of the 12 HRQoL parameters.
	Probert et al. [28]	EQ-5D-3L	115 UC	No	The combined (oral + rectal) therapy group reported a significant improvement in the ‘mobility’, ‘usual activity’ and ‘anxiety/depression’ domains at week 4.
Thiopurines	Alruthia et al. [32]	EQ-5D-3L EQ-5D-VAS	160 IBD (56% CD, 44% UC)	Yes	Patients on AZA presented higher HRQoL at six-month follow-up compared with patients on other treatments (β = 9.35; 95% CI: 0.486–18.22; *p* = 0.003).
	Bastida et al. [49]	SF-36 IBDQ	92 IBD (68 CD, 24 UC)	Yes	Compared with baseline, 68 and 64% patients’ scores improved at 6 and 12 months, respectively (ΔIBDQ was 0.86 and 1.05, respectively). SF-36 showed a similar improvement.
	Calvet et al. [50]	SF-36	33 RCD ^a^, 14 ACD ^b^, 66 HC ^c^	Yes	SF-36 were 85 in RCD, 85 in HC (*p* = 1), and 58.6 in ACD (*p* < 0.001 for comparison with RCD and HC).
Infliximab	Feagan et al. [33]	IBDQ SF-36	728 UC	No	IBDQ score improvement was significantly greater in the IFX 5 and 10 mg/kg groups (40 and 36, respectively *p* < 0.001) vs. placebo (28).
	Silva et al. [34]	IBDQ	31 UC	Yes	In IFX group (*n* = 21), the IBDQ scores ranges from 116.2 at baseline to 170.75 and 176.62 at week 30 and 54, respectively (*p* ≤ 0.02)
	Feagan et al. [51]	IBDQ SF-36	335 CD	No	The mean change in the IBDQ at week 54 compared to baseline was 22.1 in the 5 mg/kg and 30.2 in 10 mg/kg IFX maintenance group while it was 8.9 in the placebo group (*p* ≤ 0.05). SF-36 changed in the same line.
Adalimumab	Travis et al. [36]	SIBDQ EQ-5D-5L EQ-5D-VAS	463 UC	Yes	Significant improvements from baseline to week 26 were detected on SIBDQ (mean change 17.4) and EQ5D (index: 0.1 ± 0.2; VAS: 19.5).
	Louis et al. [56]	SIBDQ	945 CD	No	60% of IFX-naïve patients and 47% of IFX primary non-responders reported clinically significant improvements (≥9 points) on SIBDQ.
	Saro et al. [58]	IBDQ EQ-5D EQ-5D-VAS	126 CD	Yes	It has been shown a significant improvement on the EQ5D from 0.735 to 0.797, the EQ5D VAS from 50.0 to 80.0, and the IBDQ from 56.7 to 67.5 (*p* < 0.05 for all comparisons).
Golimumab	Feagan et al. [37]	IBDQ SF-36	1064 UC	No	It was determined a significantly greater improvement from baseline to week 6 in GLM vs. placebo groups in IBDQ (27.2 vs. 14.6), SF-36 PCS (4.14 vs. 2.46) and MCS (4.89 vs. 1.60, *p* < 0.01 for all comparisons).
Vedolizumab	Feagan et al. [38]	IBDQ SF-36 EQ-5D-3L ED-5D-VAS	373 UC	No	Patients on VDZ reported significantly greater improvements in IBDQ and EQ5D-VAS scores. For EQ-5D utility score, only the VDZ every 4 weeks group showed a significant difference from placebo. At week 52, more patients on VDZ met the minimal clinically meaningful difference thresholds for IBDQ, SF-36 physical component and EQ5D-VAS scores.
	Loftus et al. [39]	IBDQ	769 UC (383 VDZ, 386 ADA)	No	At week 52, clinically important IBDQ improvement was detected in a greater proportion of VDZ treated patients compared with ADA treated ones (52.0% vs. 42.2%). Likewise, 50.1% (VDZ) vs. 40.4% (ADA) of patients achieved IBDQ remission.
	Vermiere et al. [59]	IBDQ EQ5D-VAS SF-36	1349 CD	No	At week 80, the mean changes from baseline HRQL scores were >51 for IBDQ, >23 for EQ-5D VAS, >9 for SF-36 PCS and >10 for SF-36 MCS.
	Parkes et al. [60]	SIBDQ	61 IBD (21 CD, 40 UC)	No	SIBDQ score increased by 8.5 and 10.2 points in CD and UC patients, respectively, at week 14.
	Eriksson et al. [61]	SHS	169 CD	No	It has been seen a significant decreased of the SHS score at week 52 (*n* = 68; *p* < 0.001)
Ustekinumab	Sandborn et al. [41]	IBDQ SF-36	284 UC	No	55.6% of patients who had been treated with USK were in IBDQ remission. Regarding the SF-36, 50.0% and 45.1% of patients had a clinically meaningful improvement in the PCS and the MCS, respectively.
	Sands et al. [62]	IBDQ SF-36	1368 CD	No	A clinically meaningful improvement in IBDQ score at week 8 was achieved in 68.1% of anti-TNF naïve patients and 54.8% of patients with previous failure to antiTNF. Similarly, greater improvements in SF-36 in the USK group have been determined.
	Marquès et al. [63]	IBDQ	33 CD	Yes	18% achieved IBDQ normalization at week 52.
Tofacitinib	Panés et al. [42]	IBDQ SF-36	1161 UC (induction) 593 UC (sustain)	No	In OCTAVE induction 1 and 2, mean IBDQ changes from baseline to week 8 was 40.7 and 44.6 with TFC 10mg twice daily versus 21 and 25 with placebo, respectively (*p* < 0.001). Mean SF-36 changes were comparable with the IBDQ changes and both were sustained at week 52
Surgery (intestinal resection)	Wright et al. [64]	IBDQ SF-36	174 CD	No	A significant improvement has been observed at 6 months postoperatively compared to preoperatively in PCS (68 vs. 40), MCS (68 vs. 44) and IBDQ (171 vs. 125; *p* < 0.001 for all comparisons).
	Ha et al. [65]	5 generic tools 3 disease-specific tools	1108 CD	Yes	Both generic and disease-specific tools showed an improvement in HRQoL from 2 weeks after intestinal resection for up to 5 years.

AZA: azathioprine. IFX: infliximab. GLM: golimumab. VDZ: vedolizumab. ADA: adalimumab. USK: ustekinumab. TFC: tofacitinib.^a^ Remission Crohn’s disease. ^b^ Active Crohn’s disease. ^c^ Healthy controls.

The LIRIC study (a randomized trial that compared laparoscopic ileocecal resection with infliximab in patients with CD) showed similar results in terms of HRQoL in both groups. One hundred forty-three patients with inflammatory ileocecal CD who had previously failed conventional treatment were randomized to infliximab or ileocecal resection. The mean IBDQ score at 1 year was 178.1 (95% CI 171.1–185) in the surgery group and 172 (96% CI 164.3–179.6) in the infliximab group. The authors concluded that laparoscopic resection had similar HRQoL outcomes to treatment with infliximab and could be a reasonable option in this group of patients [68].

## 5. Why should HRQoL Be a Target in IBD?

There are scarce studies that have studied the relationship between potential targets in IBD and HRQoL. In a multicenter study performed in Spain which included 115 IBD patients, it was observed that among patients who achieved mucosal healing, 82% in CD and 78% in UC could normalize their HRQoL [69].

Psychological aspects related to chronic conditions, like anxiety and depression, have a close relationship with HRQoL [70]. In a cross-sectional prospective study performed in 875 consecutive IBD patients, all completed the Hospital Anxiety and Depression Scale (HADS) questionnaire, the Perceived Stress Scale (PSS) questionnaire and the COPE questionnaire to measure psychological alteration. In order to assess HRQoL, the SF-36 and the IBDQ-36 questionnaires were also completed. Authors concluded that high levels of anxiety, depression and stress were found to be associated with low levels in all quality of life measurements [25]. These results made some authors think about the need not only to have the physical component of HRQoL as an endpoint in IBD, but to go further and also consider psychological remission as a future endpoint in IBD too [71].

The importance of HRQoL as endpoint for new drugs is going to increase in the coming years. There is no doubt that for IBD patients it is essential to normalize their HRQoL for their daily life. We as physicians have to realize that patients with normal HRQoL are going to have a better prognosis and evolution of their disease.

## Figures and Tables

**Table 1 ijerph-18-07159-t001:** Characteristics of tools for measure HRQoL in IBD.

		Target	Recall Period	Number of Items	Response Options	Range of Scores (Worst-Best)	Reliability
Specific-disease tools	IBDQ-32	IBD	2 weeks	32	7-Level Likert (1–7)	32–224	+++
	SIBDQ	IBD	2 weeks	10	7-Level Likert (1–7)	10–70	++
	IBDQ-36	IBD	2 weeks	36	7-Level Likert (1–7)	36–252	NA
	IBDQ-9	IBD	2 weeks	9	7-Level Likert (1–7)	0–100	++
	CUCQ-8	IBD	2 weeks	8	4-Level Likert (0–3) or ordinal format (0–14)	90–0	+++
	CLIQ	CD	Today	27	True/Not true (1–0)	27–0	+++
	IBDQ-D	UC-IPAA	2 weeks	32	7-Level Likert (1–7)	32–224	NA
	CAF-QoL	CD	6–8 weeks	28	4-Level Likert (0–4)	112–0	+++
Generic tools	SF-36	Patients and general population	4 weeks	36	Linear transformation of raw scores	0–100	+++
	EQ-5D	Patients and general population	Today	6	− 3-Likert (1–3) − 5-Likert (1–5) − visual analogue scale	243 health status, index 0–1 − 0–100	+++

+ Poor, ++ Fair, +++ Good.

## Data Availability

Not applicable.

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
