# Peer review of "Role of Quality of Life as Endpoint for Inflammatory Bowel Disease Treatment"

_ijerph, 2021, doi:10.3390/ijerph18137159_

Round 1

Reviewer 1 Report

Dear editor, 

I read this review article with much interest and the article is well written. the authors have referenced several significant and land mark studies that have defined a successful treatment of IBD patients through health related quality of life measures. Overall, the article flow is very good. And the conclusions are appropriate to highlight the importance of quality of life scores in managing IBD patients. 

Minor comments: 

There are several grammatical errors that needs to be fixed. some statements need to be re-written. I have highlighted several of them. But would suggest to revise the article and get it proof read it by a native English speaker. 

As the authors have mentioned, there is need for screening IBD patients for anxiety, depression, and IBS symptoms. This was noted in Nazarian et. al., study PMID: 33855267as well.  

Author Response

Thank you for your help and kind words, we added the mentioned reference and we reviewed the writing with a native English speaker as you suggested.

We have also added Nazarian et al article in references

Reviewer 2 Report

Authors Cristina Calviño-Suarez, Rocio Ferreiro-Iglesias, Iria Baston-Rey, Manuel Barreiro-de Acosta in their work entitled "Role of quality of life as  endpoint for inflammatory bowel disease treatment" presented a very important aspect concerning the health-related quality of life (HRQoL) that has been defined as the value assigned to the duration of life influenced by physical and mental health. Considering that IBD is a chronic disease affecting especially young people, characterized by periods of remission and exacerbation. These diseases require chronic pharmacotherapy, diet therapy, and sometimes surgical treatment and the associated consequences, such as SBS- Short Bowel Syndrome. For years, doctors have been trying to bring their patients into a period of clinical and endoscopic remission - mucosal healing. As it is known, the effect of treating patients with CD and UC depends on the drugs or used surgical techniques. The aspect of the quality of life as one of the endpoints for new drugs used for IBD treatments becomes very important.

Thus, HRQoL has been included as an outcome in numerous studies evaluating different IBD therapies, including clinical trials but also real-life studies

The authors summarized the literature to date on the impact of the applied therapy on the HRQoL of IBD patients in a very orderly manner.
The work is written in an interesting way, in a good understandable language, and takes into account most of the current news.

I would suggest that authors carefully check the grammatical as well as spelling site, as there are a few mistakes.   

Author Response

Thank you for your kind report, we finally reviewed the writing with a native English speaker.

Reviewer 3 Report

Dear Authirs,

I have several concern regarding your manuscript:

General remark: the manuscript should be organized in more comprehensive way-it is difficult to follow. Quality of Life described in CD and UC patients can be presented in common table and in the text only the most important details information should be highlighted. The information about several tools used for QoL assessment can also be presented in table, especially information on accuracy or precision. 

I do not have a feeling that Authors structured the manuscript according some main idea. This should be changed before next round of revision.

Author Response

Thank you for your comments. We structured the manuscript in 5 main points following a scheme which starts with the HRQoL concept and how it recently became more important in IBD, which are the most used tools to assess it and how can we manage it in our patients given the evidence of the effect of IBD treatments on it. We made a table including the most important studies that demonstrated which IBD treatments improve HRQoL, including Crohn’s Disease and Ulcerative Colitis. Finally, on the table which summarizes some tools to assess HRQoL, we added a column to share the precision or reliability of each.